# Neural Representations in Multi-Task Learning guided by Task-Dependent Contexts

## Abstract

The ability to switch between tasks effectively in response to external stimuli is a hallmark of cognitive control. Our brain is able to filter and integrate external information to accomplish goal-directed behavior. Task switching occurs rapidly and efficiently, allowing us to perform multiple tasks with ease. In a similar way, artificial neural networks can be tailored to exhibit multi-task capabilities and achieve high performance across domains. In terms of explainability, understanding how neural networks make predictions is crucial in many real-world applications, for instance, in guiding clinical decisions.

In this study, we delve into neural representations learned by multi-tasking architectures. Concretely, we compare *individual* and *parallel* networks with *task-switching* networks. *Task-switching* networks leverage task-dependent contexts to learn disentangled representations without hurting the overall task accuracy. We show that *task-switching* networks operate in an intermediate regime between *individual* and *parallel* networks. In addition, we show that shared representations are produced by the emergence neurons encoding multiple tasks. Furthermore, we study the role of contexts across network processing and show its role at aligning the task with the relevant features. Finally, we investigate how the magnitude of contexts affects the performance in *task-switching* networks.

## 1 Introduction

Living involves constantly gauging and selecting the optimal task to perform. This decision is the result of the interaction of different elements, such as our current goals, external circumstances, or the stimulus context (Monsell, 2003). In the brain, task switching and cognitive control has been associated principally with the prefrontal cortex (PFC), which provides top-down regulation to other cortical brain areas Johnston et al. (2007). In this sense, PFC controls the activation of multiple neural pathways that are activated or deactivated to ultimately result in task execution. Neural pathways are formed by collections of neurons that cooperate to yield a specific effect. Although single neurons have been historically regarded as centre of processing in the brain, now we are moving towards a framework in which neural networks conform the functional processing unit of the nervous system Yuste (2015). In the visual cortex, where subpopulations of neurons are responsible of the processing of different visual field features for correct object recognition DiCarlo et al. (2012).

According to the gating theory (Miller et al., 2001), context-dependent top-down inputs from the PFC regulate the activation of neural pathways. The encoding of relationships between stimulus and contexts can be facilitated by tuning neural activity to multiple tasks. In the PFC of monkeys, it has been observed that single neurons exhibit nonlinear responses to multiple stimuli (Rigotti et al., 2013). This behavior, or mixed selectivity, favors high-dimensional representation of neural activation, which allows linear readouts to generate a vast number of responses, in opposition to low-dimensional representations (Fusi et al., 2016).

Recently, artificial neural networks have been revisited as models of neural computation, many findings suggesting their practicality for assessing brain theories Richards et al. (2019). For example, the internal representations learned by neural networks have been associated with representations in the brain in multi-tasking settings (Ito et al., 2022; Ito & Murray, 2021; Flesch et al., 2022a;b).

In this paper, we investigate the neural representations learned by feedforward multi-tasking architectures. Neural networks have been designed to be capable of processing multiple tasks in parallel,

which is beneficial for the network performance (Caruana, 1997) to achieve high performance across domains Ruder (2017). Here, we focus on neural networks using contexts to switch attention between tasks. We use population analysis tools to investigate how neural computations are associated with task stimulus Kriegeskorte et al. (2008); Jazayeri & Ostojic (2021) and describe the advantages of learning representations using task-switching networks.

## 1.1 RELATED WORK

Neural networks with contexts have been used as models to study cognitive control in machines and humans. Mante et al. (2013) reproduced PFC dynamics from monkeys using recurrent neural networks with sensory contexts. Similarly, Ardid & Wang (2013) analyzed task-switching behavior effects, such as switch and congruency, emerging from network attractor dynamics. In Musslick et al. (2017), authors studied the learning efficiency of neural networks in multi-tasking with contexts and tasks with ranging degrees of overlapping. Flesch et al. (2022a) analyzed the geometry of representations learned by neural networks and humans in a task-switching schedule. Later, Flesch et al. (2022b), modified the stochastic gradient descent algorithm to strengthen task-relevant features in a continual learning setting. More recently, Ito et al. (2022) studied the generalization of new tasks by composing old tasks using different rule-based contexts. In Ito & Murray (2021), authors used a neural network to investigate the transformation mapping between visual and motor representations occurring in the brain. They used representational similarity analysis to study the geometry of neural codes in multi-tasking Kriegeskorte et al. (2008); Kriegeskorte & Kievit (2013).

Contexts have been used to alleviate catastrophic forgetting in continual learning. Masse et al. (2018) showed that adding context to parameter stabilization methods improved accuracy when compared to parameter stabilization alone. Serra et al. (2018) implemented gating by using task-based attention mechanisms to help preserving information of previous tasks without compromising learning new tasks. In Grewal et al. (2021), authors combined synaptic intelligence with active dendrites and sparse activations to reduce catastrophic forgetting. The idea of recruiting multiple subnetworks for continual learning was previously explored in Wortsman et al. (2020).

In addition, Li et al. (2016) proposed multi-bias non-linear activations to improve feature extraction in convolutional neural networks, and for semantic segmentation, task-switching networks using task embeddings were introduced by (Sun et al., 2021) to promote learning common parameters across the tasks using the same network.

## 1.2 OUR CONTRIBUTIONS

The main contributions of this paper are two-fold:

1. Firstly, we investigate the representations learned by three different variations of feedforward networks. We find that task-switching network operate in an intermediate regime between individual and parallel networks and the performance on all tasks is improved when parameter sharing is present (section 3.1).

2. Secondly, we expand previous analyses involving multi-task learning with contexts and mixed selectivity (section 3.3) and report new findings on the impact of contexts location and magnitude at different stages of processing (sections 3.2, 3.4, 3.5).

## 2 METHODS

### 2.1 ARCHITECTURES

We conducted different experiments with three variants of the feedforward network (Rumelhart et al., 1986), (Goodfellow et al., 2016) in multi-task learning:

1. **Individual Networks**: Each task is learned by an independent network. Multi-tasking is performed by combining the outputs of the networks. An individual network is parameterized as $\mathbf{y_t} = f_t(\mathbf{x}; \theta_t)$, where $t \in T$ denotes the task in the set $T = \{T_A, T_B, ..., T_N\}$ of $N$ tasks, $\theta_t$ denotes the weights and biases specific for each task, $\mathbf{W_t}$ and $\mathbf{b_t}$ for each layer. Parameters here are independent, hence not shared across tasks.

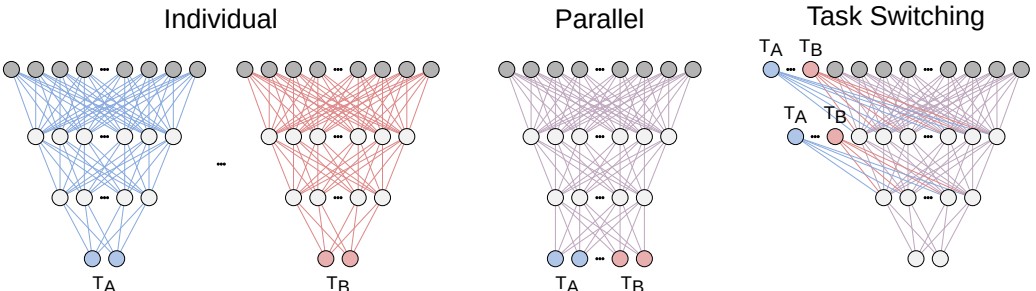

Figure 1: Architectures for multi-task learning addressed in this paper.

2. **Parallel Networks**: Tasks are processed simultaneously and the output layer accounts for independent responses for each of the tasks. The parallel processing regularizes the network and allows overall better performance than the individual networks Caruana (1997). A parallel network is defined as $\mathbf{y} = f_t(\mathbf{x}; \theta)$. The parameters in hidden layers are exactly the same for both tasks, hence fully shared. The only independent parameters in this network are the ones present in the output layer.

3. **Task-switching Networks**: Each task is processed individually, but parameters, other than the context biases, are fully shared. The output of a task-switching network is $\mathbf{y_t} = f_t(\mathbf{x}; \mathbf{W}, \mathbf{b_t})$. Weights are co-optimized for all tasks, but bias vectors are different. We define a task-switching network to be a composition of layers, which can the context bias for each task, $y = \mathbf{W^T}\mathbf{x} + \mathbf{b_t}$ or not $y = \mathbf{W^T}\mathbf{x}$. A task-switching network requires at least one context layer with bias. We train task-switching networks following the interleaved task-switching schedule (Flesch et al., 2018).

Architectures are depicted in figure 1. The detailed description of the training conditions are described in section A.1.

## 2.2 TASKS

In this study, we created five *binary tasks* using the MNIST database (LeCun et al., 1998). The main task pair we use throughout the study corresponds to *parity* and *value*. Parity defines a digit as either even or odd, whereas value defines a digit being smaller or equal to four, or larger. We analyzed the impact of *congruency* in learning representations. For two different tasks, a stimulus (digit) is *congruent* if the output labels for both are the same. The opposite is referred as *incongruency*. The detailed list of tasks and outputs is explained in section A.2.

## 3 RESULTS AND DISCUSSION

### 3.1 TASK-SWITCHING NETWORKS OPERATE IN AN INTERMEDIATE REGIME BETWEEN INDIVIDUAL AND PARALLEL NETWORKS

The first experiment consisted in analyzing the neural representations of three different multi-task learning networks. Similarly to (Caruana, 1997), we compared multi-tasking using individual neural networks and parallel neural networks. In addition, we included task-switching networks, where tasks are informed by binary context inputs.

To analyze the representations learned by each network, we used the representational similarity analysis (RSA) Kriegeskorte et al. (2008). We build the representational dissimilarity matrix (RDM) by grouping, in terms of digit images, the neural activations of each layer. For each digit, we averaged the neural activity and computed their dissimilarity. We repeated this process for different tasks and calculated the dissimilarity between class labels (digits). Finally, we assembled the RDM. Section A.3 addresses the calculation of the RDM in detail.

For individual and task-switching networks, calculating the dissimilarity between tasks can be done directly. For individual networks, we have specific activity patterns from each network performing

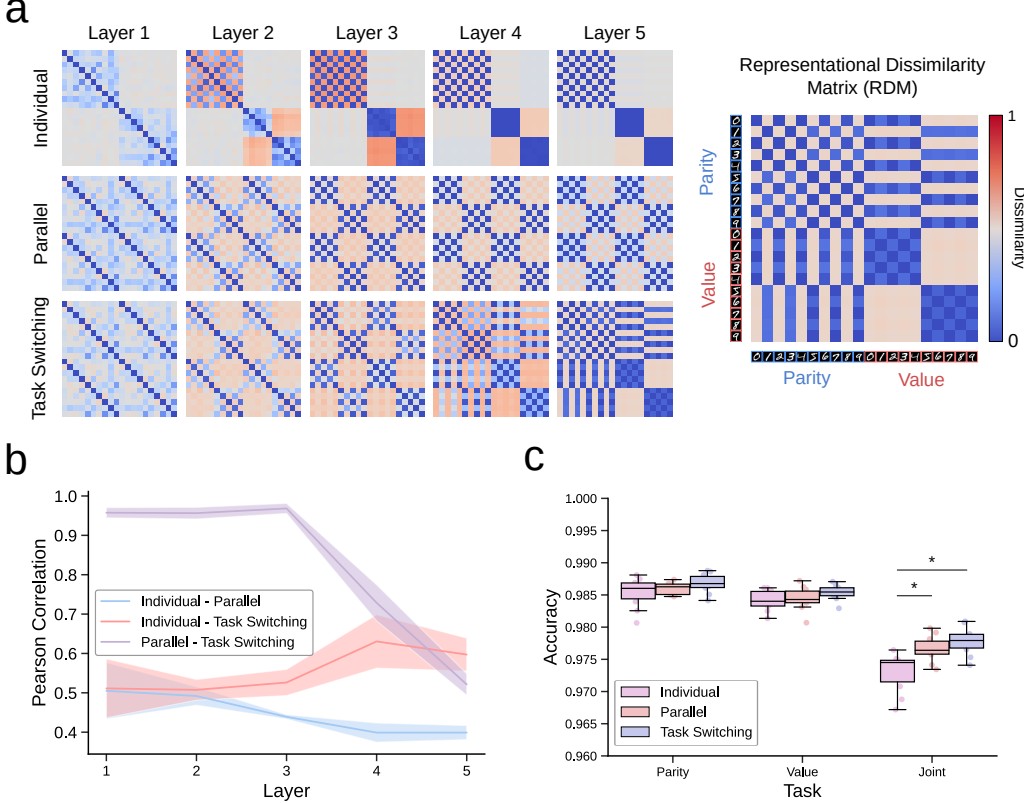

Figure 2: (a) Average RDM for the *parity/value* task-switching schedule for the three architectures for 10 different runs. (b) Pearson correlation coefficient between RDMs. (c) Accuracy of the different models.

each task, and for task-switching networks, we have specific activity patterns originated from the same network under each task context. For parallel networks, since the internal processing for both tasks is exactly the same, the resulting neural representations are entangled. Then, we construct the RDM by simply repeating the same activation patterns.

Figure 2a shows the average RDM for the three models after training with *parity* and *value*. Individual networks do not share representations between tasks (off-diagonal). Parallel networks separate partially the congruent and incongruent digits to resolves the active task in the output layer. Task-switching networks behave similarly to parallel networks in the first layers but resolves the task before the output layer, similarly as the individual networks. Importantly, we can reconstruct the task goal from the activity of both individual and task-switching networks. However, we can only reconstruct the task congruency from parallel networks. Additionally, in comparison to individual networks, task-switching networks share representations between tasks. To quantify the degree of similarity between representations, we calculated the Pearson correlation coefficient between RDMs for different runs. Results in figure 2b reinforce the previous assessment: task-switching networks behave similarly to parallel networks in the first stage of processing, but ends significantly more similar to the individual networks.

Finally, we assessed the performance of networks and analyzed the performance in terms of the accuracy achieved across runs for both tasks. Figure 2c shows a similar performance of the three networks for the two tasks, yet significant differences appeared between individual and parallel networks, as well as between individual and task-switching networks, when performance was evaluated simultaneously for the two tasks (Wilcoxon Signed Rank Test, $p < 0.05$). In contrast, parameter sharing between tasks in parallel and task-switching networks regularizes the network performance.

We conducted the same analysis of figure 2 using 5 different tasks. The results are shown in figure 7.

## 3.2 TASK-SWITCHING NETWORKS PERFORM MULTIPLE TASKS BY EXTRACTING TASK CONGRUENCY INFORMATION

The second experiment consisted in analyzing the representations learned by task-switching networks when contexts were removed after training. Tasks share a common pool of neural activations that is biased towards resolving one task or another depending on the context. To study this common computation, first we investigate the RSA by calculating the correlation between RDMs of three variants of the task-switching network: only context in the first layer (*first*), context in all layers (*all*), and trained with all contexts but removed at inference time (*removed*). Figure 3a shows the RSA between these networks. The *removed* and *all* networks have similar representations in the first layers, only to diverge as we approach the output layer. Contrarily, *first* and *all* start with lower correlation but by layer 5 they reach maximum correlation. The drop of correlation in the first layers is caused by the interference of the only context with the feature extraction of digits in the *first* network.

To further analyze the representations, we computed multidimensional scaling (MDS) on the RDMs (Jazayeri & Ostojic, 2021). For implementations details see section A.4. Figure 3b shows the MDS for the three first layers and the last two. Digits are represented differently for both tasks in the *first* network, since the context bias has been introduced early in the processing. Here, the network is forced to resolve the task before the context input dissapears. In layer 3 of the *all* and *removed* networks, we see the projection of digits into almost orthogonal dimensions: one dimension represents incongruent stimuli, whereas the other represents congruent stimuli.

In addition to MDS, we used a generalized linear model to decode the activations in the original dimensional space (see section A.5). Figure 3d shows the performance of the classifier when we tested the linearity of digit, task and congruency. In high-dimensional space, digit is decoded with high accuracy in the first layers. For all three models, the accuracy collapses in the last layers. The *removed* and *all* networks are slightly better than *first*, possibly due to the context interference in the first layer. Task information is already present in the first layer of the *first* network. The *all* network shows low accuracy in the first layers, as if the contexts where not acting before the third layer (more about weight importance in section 3.4). Finally, to test the hypothesis that task-switching networks extract congruency information, we evaluated the performance on congruency itself. The *removed* network is capable of decoding perfectly congruency across layers in the high-dimensional space defined by the neural activations. Here, the *first* network shows the greatest decrease in accuracy when approaching the last layers. This plot confirms the hypothesis that contexts bias neural activity towards the goal of the task. During training, both tasks are being optimized together, reaching some local minima that reduces the loss for both tasks. Congruency is the midpoint between the optimization of both task and determines the amount of shared representations. In the extreme case of two identical tasks, the overlapping is maximum and the bias signals would converge, reducing then the multi-tasking problem to a single-task problem, with contexts working as regular activation biases.

## 3.3 MIXED SELECTIVITY EMERGES FOR RESOLVING TASK EXECUTION

After having inspected the behavior of neural populations in single layers, we wanted to investigate the role of single units, specially in the last layer, where the network needs to predict the appropriate output. We hypothesized that neurons have to organize to drive collectively the dynamics towards the task goal. Mixed selectivity has been previously observed in artificial neural networks with a single hidden layer and contexts in the first layer (Flesch et al., 2022a). Here, we analyzed the activations of neurons in last layer and performed agglomerative hierarchical to cluster neural activity. The method employed is detailed in section A.6.

The results of clustering are shown in figure 4. The different plots represent the mean activity of the different clusters for each task when presented with one of the digits. We used two networks from the previous section, *all* and *removed*. The activity was normalized to a maximum of 1 (radius of the plot). We observed the existence of two different clustering patterns. In both, we found a cluster of silent units that is not active for any of the numbers. The first row of the plot represents

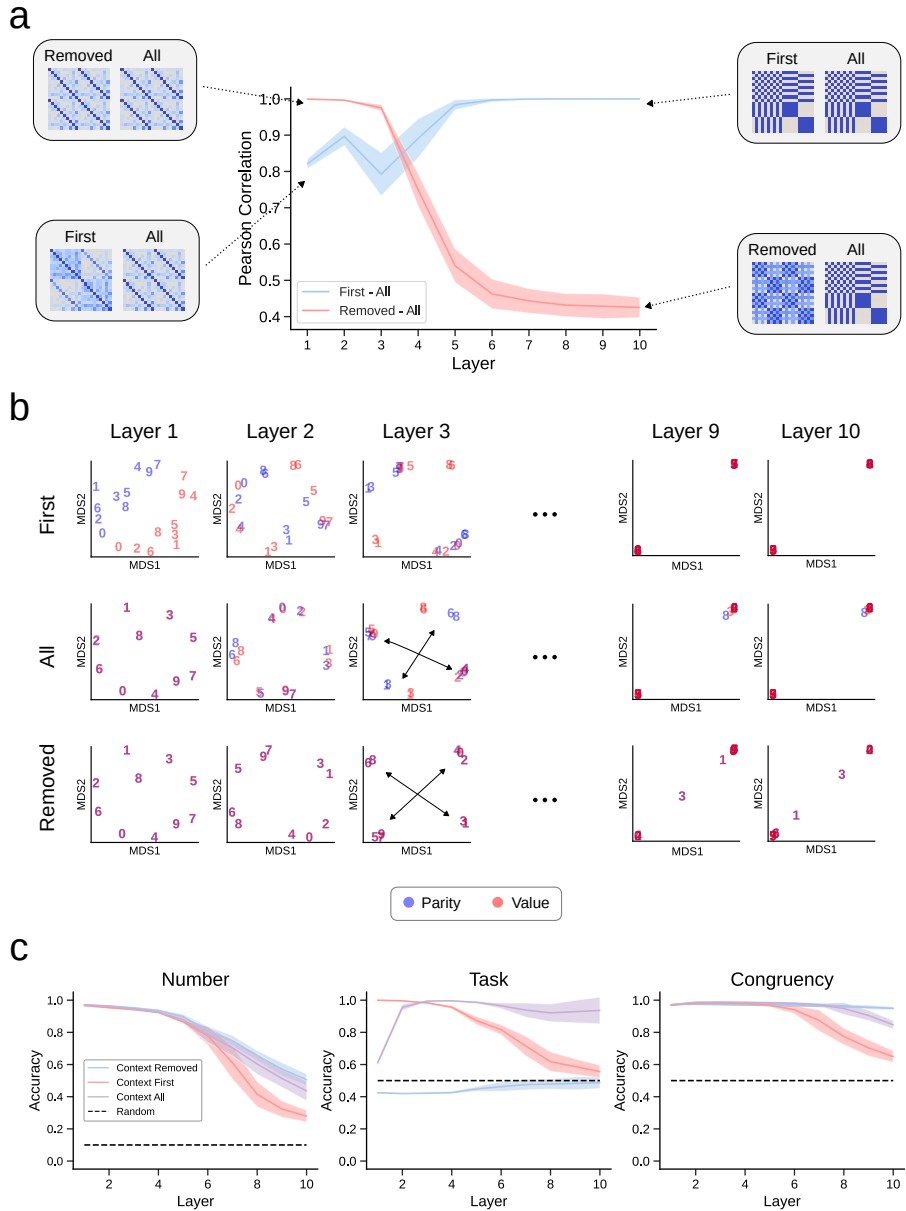

Figure 3: (a) Mean and standard deviation of RSA between *first*, *all* and *removed* network for 10 different runs. (b) Multidimensional scaling (MDS) of the representational dissimilarity matrix (RDM). (c) Mean accuracy and standard deviation of the linear decoder of neural activity for 10 different runs.

the first pattern. In this pattern of activation, there is a single cluster that responds to even digits in parity and values less than or equal to four in the value task. When contexts are removed the cluster responded only to the even digits and values less than or equal to four, entailing that context increases the activity for 6 and 8 in parity and 1 and 3 in value, that is, for the incongruent digits. For the other digits, the network thresholds neural activity. For example, odd digits in the parity task show silent activity.

The second pattern of activation is showed in the second row of figure 4. We found two main clusters of activity. The first cluster accounts for even digits and values less than or equal to four, the second cluster for odd digits and values greater than four. When contexts were removed, clusters only encoded the congruent digits. These patterns of activations show how neurons are tuned to

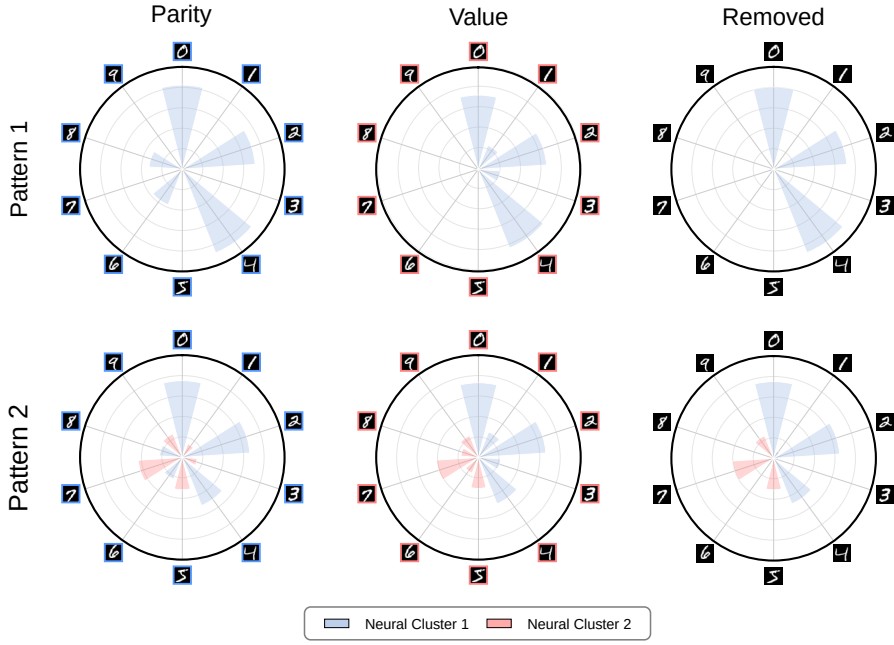

Figure 4: Neural clustering for the *parity/value* task-switching schedule.

different tasks depending on the current active stimulus. In the brain, neurons that are responsive to multiple tasks are said to exhibit mixed selectivity (Rigotti et al., 2013). We repeated the experiment using five tasks and found that the majority of patterns resembled pattern number two. These results appear in figure 7 in the appendix.

## 3.4 CONTEXT STRENGTH IS MAXIMIZED AT INTERMEDIATE LAYERS

After observing the impact of context in the first layer in the previous section, we hypothesized that context location could be essential for proper input feature extraction and to drive the activity towards the task goal. Flesch et al. (2022a) observed anticorrelation in the contexts of tasks situated in the first layer in networks with one hidden layer. Here, we run a series of experiments to assess the strength of contexts varying their position across the network. Figure 5 shows the results. Each row represents a different context configuration and each column is the index of the layer (we fixed the maximum number of layers to 10). A cell colored according to the heatmap means that the network has a context at that position. A gray cell means that the network has a layer but no context. A white cell denotes the absence of the layer, and thus the network is shallower. We tried four different settings: adding contexts from the first layer (5a), adding contexts starting from the last layer (5b), adding a single context at different layers (5c) and adding contexts and layers at the same time (5d). In the two first heatmaps we show the Frobenius norm of the weights associated with the corresponding task. Then we calculated the correlation between contexts of different tasks at the same location. Finally, we show the average accuracy (and standard error) of the networks in the same row. The dashed line in the accuracy plot represents the mean accuracy achieved by a network of 10 layers and 10 contexts (control case).

Results show that context location is not trivial. Largest context strength is achieved at intermediate layers, where the norm of contexts is higher. At the same time, the anticorrelation between contexts of different tasks is higher. Context have positive correlation if located at the end of networks with multiple contexts. In addition, adding context at the beginning or at the end of networks are associated with the largest magnitude of context strength. It is explained by the fact that the network is constrained to digit processing and task separation in the former case, and task separation and output mapping in the latter. When assessing the accuracy, the presence of contexts in intermediate layers achieved the same overall performance to that of the control case. Finally, we compared the biases

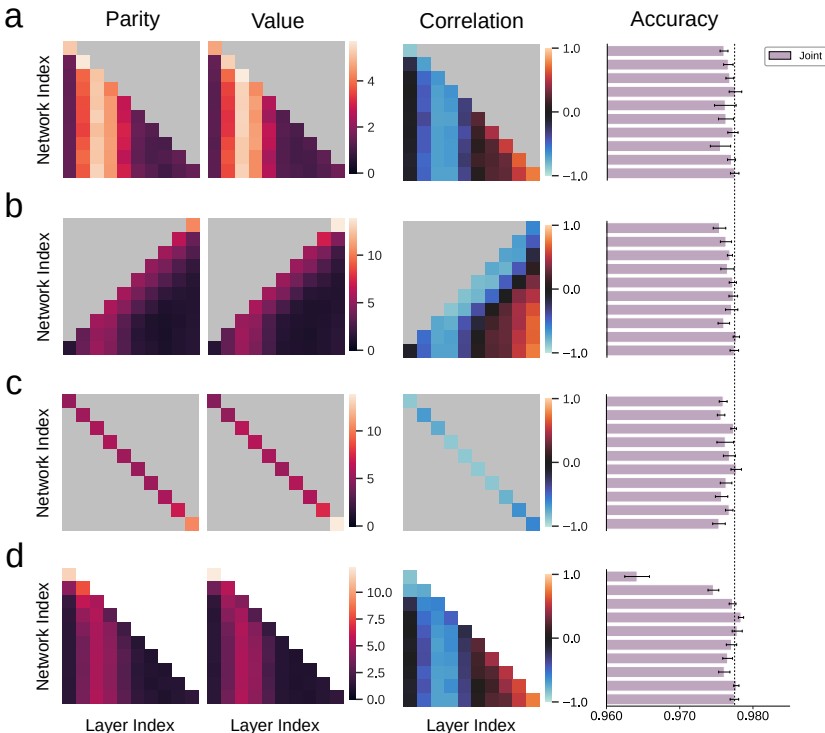

Figure 5: Mean magnitude and correlation between contexts using different network architectures for 10 different runs. (a) Context are added from the first layer. (b) Context are added from the last layer. (c) One context per network. (d) Networks have different number of layers.

between individual and task-switching in figure 9 and show that large magnitude at intermediate layers is exclusive for task-switching networks.

### 3.5 TASK-SWITCHING NETWORKS GENERALIZE REPRESENTATIONS BETWEEN TASKS

The last analysis involved the study of the neural representations when the magnitude of contexts was modified from the values of training. For each task, since we use binary (one-hot) encoding, the contexts have a default value of 1 if the task is to be executed and 0 otherwise. Values between 0 and 1 can be for instance related to task uncertainty, either normalized to sum 1 or independently from one another. Hence, many contexts could be activated at the same time. We used a 10 layer network with contexts in all layers. In a last experiment, we analyze the behavior of the task-switching network against abnormal context values ($>1$), which interest is to analyze task performance under aberrant context modulations, such as those that could be originated from abnormal activity in PFC. In these scenarios, we inspect the neural representations to gain insight of how networks perform tasks.

First, we studied the how is the transition from one task to another. To do so, we interpolated the values of contexts between the two tasks, passing by an intermediate value of 0.5. The total sum of the contexts was kept to 1, and values are complimentary. As mentioned earlier, this could be related to task uncertainty in a normalized fashion. Figure 6 shows the accuracy for the different values of the context. We moved from task *value* to task *parity*. Results show that the transition between tasks is smooth: there is not a hard threshold at 0.5, but a progressive change from one task to the other. The neural representations at 0.5 show an intermediate state between tasks. The baseline accuracy is 0.6 since it is the level of congruency between tasks. The accuracy at 0.5 is close to 0.8, which is the random chance once the congruency baseline is considered.

Next, we wanted the strength of opposition between tasks. We incremented the value of both contexts simultaneously from 0 to 1. This is a way to parametrically analize performance according

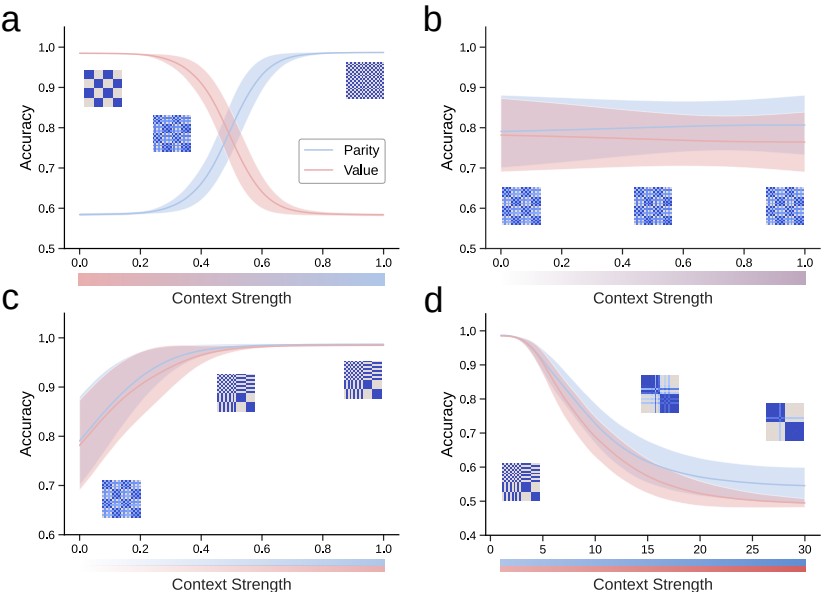

Figure 6: Mean accuracy of tasks as functions of context intensities for 10 different runs. Shaded area denotes standard deviation. (a) The total context strength is kept fixed between tasks. (b) Both tasks are increased simultaneously. (c, d) Contexts are increased and evaluated independently (no interaction).

to context strength when both tasks are equally probable. Figure 6b shows the accuracy values for both tasks. The accuracy begins at 0.8 and slightly diverges when reaching a value of 1 for each task, which is a slight preference of the networks for the task *parity*. The internal representations however, show that the network operates in an intermediate step between tasks as observed in figure 6a when both tasks had contexts of 0.5.

Finally, we wanted to analyze how the network performed each single task separately when the values of the contexts were less or greater than one. As mentioned earlier, the former can be related to non-normalized task uncertainty, whereas the latter is useful to infer behavior against aberrant context inputs. In figure 6c, we swept each task independently from 0 to 1, and represented the values in the same plot. Results indicate that networks do not need to fully certain of the magnitude of context to operate at highest performance. Concretely, we observe that with contexts close to 0.5, the networks reach the maximum accuracy. We took each network and created the RDM. The results show that the neural representations for binary (full certain) contexts are already present for uncertain contexts if their strength is greater than 0.5. Figure 6d shows the accuracies for the independent tasks when the values of contexts are up to 30-fold of those in training. When the context dramatically increases, there is an asymmetry between digit extraction and task information that leads to a decrease of accuracy. It is like the network is very clear about the task to perform but has lost most of the information about the digit. Hence, the dissimilarity between mean activations becomes close to zero.

## 4 CONCLUSIONS

In this study, we have conducted a series of experiments with multi-task networks primarily focusing on task-switching networks, which leverage contexts to switch between tasks. We have seen that task-switching networks learn disentangled shared representations that allow to reconstruct the mapping between task and goal. Task-switching networks encode congruency to resolve multi-tasking. Contexts can either improve feature extraction or interfere with the network processing, depending on their position in the network. Finally, we have shown that task-switching networks generalize representations between tasks.

REPRODUCIBILITY STATEMENT

We attached int a ZIP file the code to train and generate the figures together with the submission of this paper.

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

# A  APPENDIX

## A.1  NETWORK TRAINING

For each architecture, we trained two different models with 5 (section 3.1) and 10 layers (sections 3.2, 3.3, 3.4, 3.5). Networks were trained for a total number of 50 epochs. The number of hidden units per layer was fixed to 100. We selected ReLU as the activation function Glorot et al. (2011) and initialized weights and biases following a uniform distribution in the range $(-1/\sqrt{N}, 1/\sqrt{N})$, where $N$ is the number of inputs to the layer. We used the Adam optimizer for training (Kingma & Ba, 2014) with learning rate 0.001, $\beta_1 = 0.9$ and $\beta_2 = 0.999$ and the cross entropy loss function. The models were implemented in PyTorch Paszke et al. (2019). We collected statistics over 10 different runs and controlled the random number generator for reproducibility and hypothesis testing. The code to reproduce the results is attached in a ZIP file with the submission of this paper.

## A.2  BINARY TASKS

We designed five main tasks using MNIST. For all tasks, we split the images into a training set of 50,000 images and a test set of 10,000. The batch size was fixed to 100 images. The following table denotes the names of the task and the expected output of the network (0, first output unit, 1 second output unit):

| Task Name | Number | | | | | | | | | |
|---|---|---|---|---|---|---|---|---|---|---|
| | 0 | 1 | 2 | 3 | 4 | 5 | 6 | 7 | 8 | 9 |
| Parity | 1 | 0 | 1 | 0 | 1 | 0 | 1 | 0 | 1 | 0 |
| Value | 1 | 1 | 1 | 1 | 1 | 0 | 0 | 0 | 0 | 0 |
| Prime | 0 | 0 | 1 | 1 | 0 | 1 | 0 | 1 | 0 | 0 |
| Fibonacci | 1 | 1 | 1 | 1 | 0 | 1 | 0 | 0 | 1 | 0 |
| Multiple 3 | 0 | 0 | 0 | 1 | 0 | 0 | 1 | 0 | 0 | 1 |

The total congruency between tasks is represented in the following table:

| | Parity | Value | Prime | Fibonacci | Multiple 3 |
|---|---|---|---|---|---|
| Parity | 1 | 0.6 | 0.3 | 0.5 | 0.4 |
| Value | 0.6 | 1 | 0.5 | 0.7 | 0.4 |
| Prime | 0.3 | 0.5 | 1 | 0.6 | 0.5 |
| Fibonacci | 0.5 | 0.7 | 0.6 | 1 | 0.3 |
| Multiple 3 | 0.4 | 0.4 | 0.5 | 0.3 | 1 |

## A.3  REPRESENTATIONAL DISSIMILARITY MATRIX (RDM)

For each layer, we construct an activation matrix consisting of $N_T \times N_{test}$ rows and $N_{hidden}$ columns, where $N_T$ is the number of tasks, $N_{test}$ is the test set size, 10,000 and $N_{hidden}$ the number of hidden units of a layer. For each label of MNIST, we will calculate the average activation for the hidden units in the layer. For each task, we create a matrix of dimension $10 \times N_{hidden}$. This matrix represent the average unit response to each of the possible stimulus of the dataset. Finally, we calculate the dissimilarity between row vectors, denoting the dissimilarity between the average of two numbers for two different tasks. We calculate the dissimilarity between vectors as $1 - r/2$, where $r$ is the Pearson correlation coefficient.

## A.4  MULTIDIMENSIONAL SCALING (MDS)

To construct the multidimensional scaling embedding we create first the RDM of a layer. The RDM is a distance matrix where a position $ij$ denotes the distance between $i$ and $j$. We project this high-dimensional points into a two-dimensional embedding using MDS. We used the scikit-learn module to implement MDS Pedregosa et al. (2011). The maximum iteration was set to 1000 to guarantee convergence. The tolerance was set to $10^{-5}$.

### A.5 Linear Decoder

We used multiclass logistic regression as linear decoder of neural activations across layers. We used the scikit-learn module to implement the logistic regression model Pedregosa et al. (2011). The maximum iteration was set to 5000 to guarantee convergence. The tolerance was set to $10^{-3}$. We used L2 regularization and cross entropy as loss function.

We constructed an activation matrix consisting of $N_T \times N_{test}$ rows and $N_{hidden}$ columns, where $N_T$ is the number of tasks, $N_{test}$ is the test set size, 10,000 and $N_{hidden}$ the number of hidden units of a layer to decode. We divided the rows into a train set and test set following a 90/10 split. We used the labels of numbers to create the task and congruency labels. We assembled the activation matrix for each layer of a network and averaged the results of the 10 different runs.

### A.6 Agglomerative Hierarchical Clustering

Similarly as in the assembling of the RDM, we average the neural activity for the 10 numbers, having a matrix of size $10 \times N_{hidden}$. Here we gather the averages of activity for the different tasks to construct a matrix of size $N_{hidden} \times 10 \cdot N_T$. Each independent row conforms the averages of activations across tasks for a single hidden unit. We normalize the activity to be maximum of 1. We apply hierarchical clustering to create the dendrogram of hidden units using the *centroid* method in SciPy (Virtanen et al., 2020). The we find the cluster of each hidden unit. Finally, we calculate the average activity of each cluster.

## B  SUPPLEMENTARY FIGURES

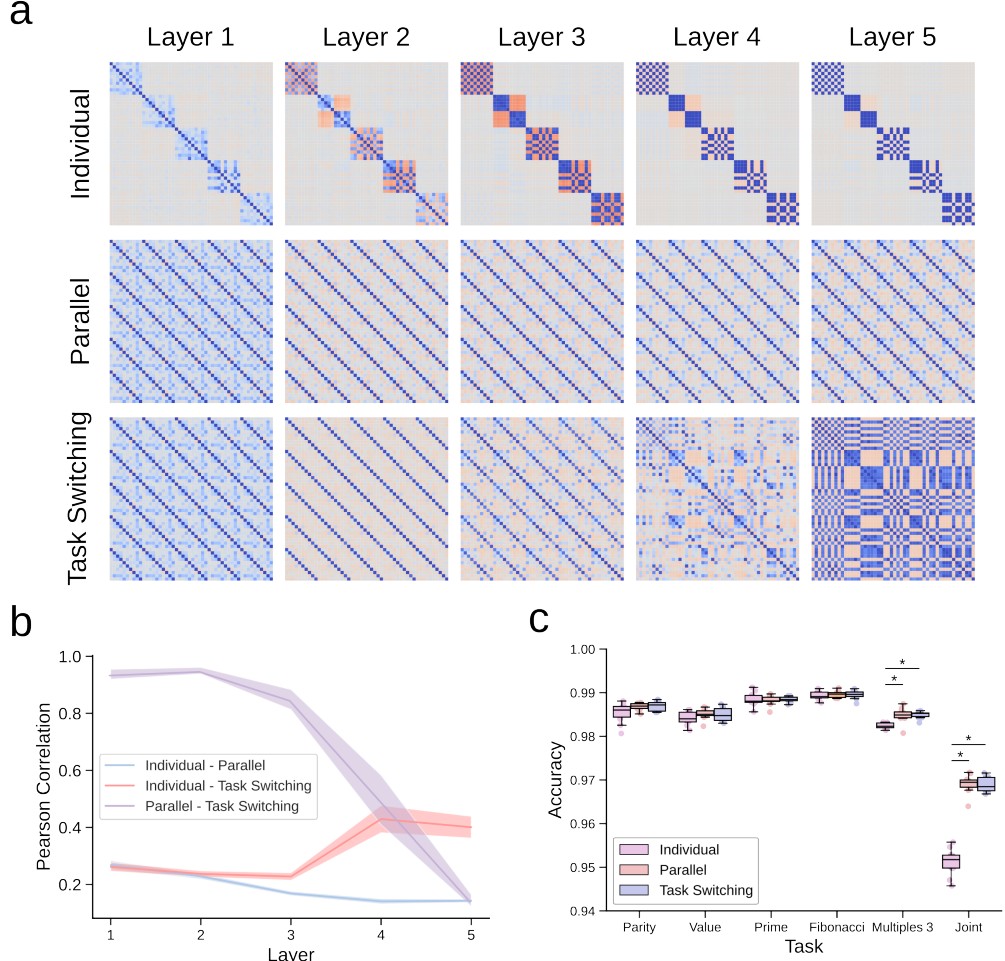

Figure 7: (a) Average RDM for the *parity/value/prime/fibonacci/multiple3* task-switching schedule for the three architectures for 10 different runs. (b) Pearson correlation coefficient between RDM (mean of 10 runs with standard deviation). (c) Accuracy of the different models.

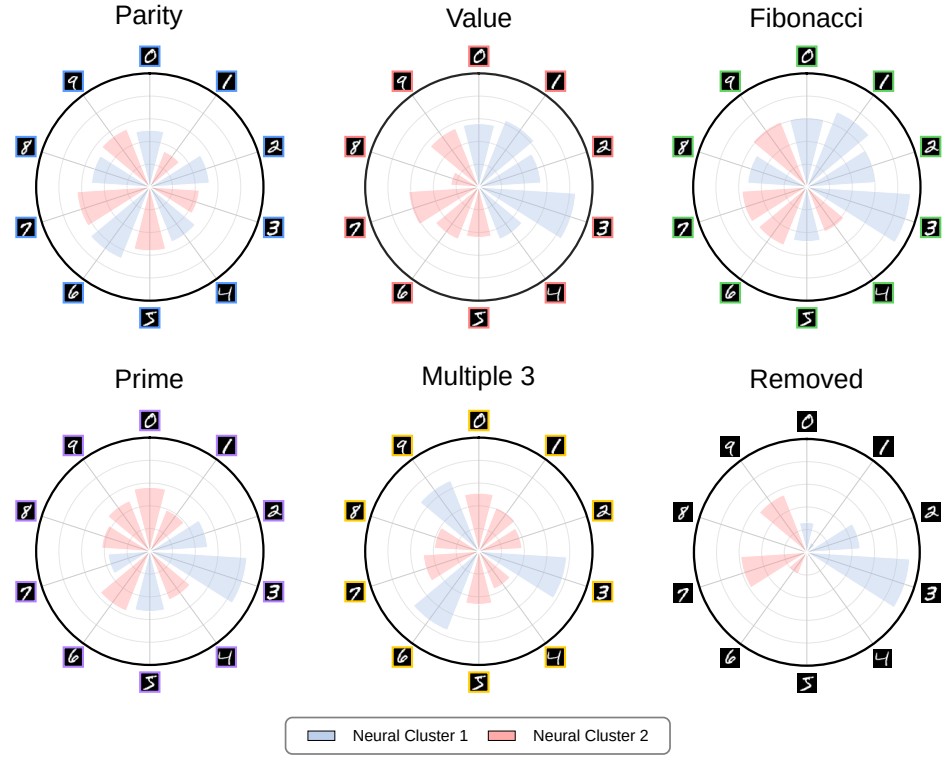

Figure 8: Neural clustering for the *parity/value/prime/fibonacci/multiple3* task-switching schedule.

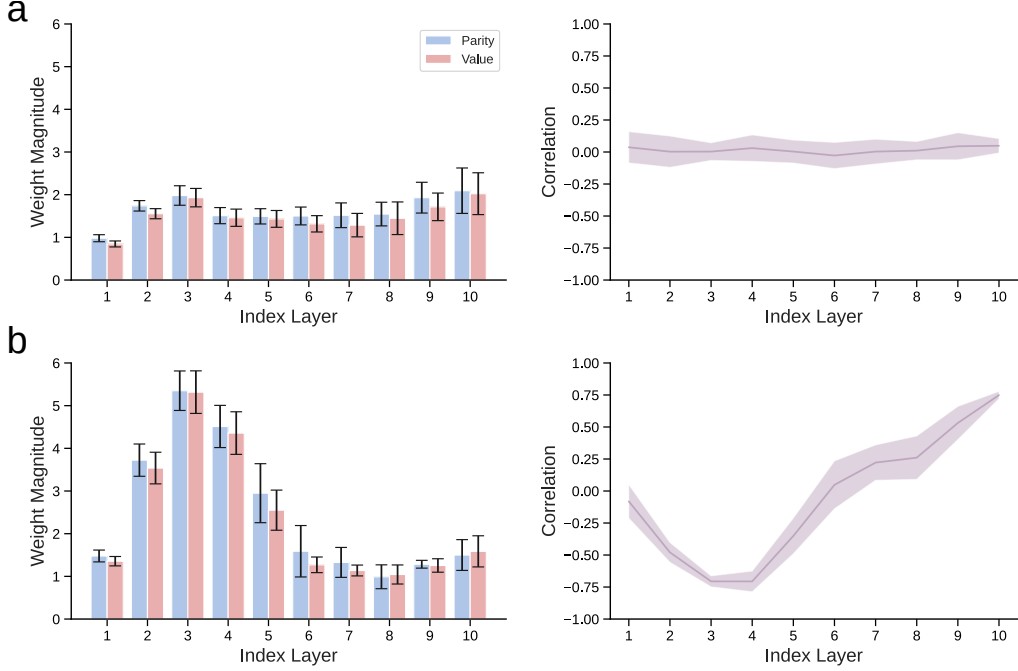

Figure 9: Magnitude and correlation between biases in individual (a) and task-switching networks (b). We show mean and standard deviation over 10 runs.

