# OpenReview forum: "Neural Representations in Multi-Task Learning guided by Task-Dependent Contexts"
_ICLR.cc/2023/Conference — Submitted to ICLR 2023_

### Official Review · Reviewer_Qimc · 2022-10-24

**Confidence:** 2
**Correctness:** 3
**Technical Novelty And Significance:** 2
**Empirical Novelty And Significance:** 2
**Recommendation:** 3

**Clarity, Quality, Novelty And Reproducibility:**

The paper is clearly written, though the authors make undue allusions between their work and multi-task handling in the human brain.

**Strength And Weaknesses:**

The work investigates an important domain, is improving the understanding of neural network representations under multi-task learning may be helpful for improving multi-task learning and explaining model predictions. It remains however unconvincing as 1) the authors do not verify if their results hold across multiple runs with different random initializations of the networks or are only random artifacts 2) MNIST is a narrow and small dataset, findings based which may not transfer to state-of-the-art applications of neural networks for example in computer vision or natural language processing which were found to benefit from multi-task learning.

**Summary Of The Paper:**

The paper investigates neural network representations under multi-task learning based on five binary classifications tasks constructed from MNIST.  The authors compare using independent, individual multi-layer perceptrons (MLPs) per task with parallel MLPs that process tasks simultaneously and with task-switching MLPs that share weights between the individually processed tasks. They show that task-switching networks learn disentangled shared representations that generalize between tasks and encode congruency.

**Summary Of The Review:**

The work is interesting but the limitations in choice of networks and datasets make it not strong enough for ICLR. Follow-up work that addresses the limitations will help to qualify the paper,

---

> ### Author Response · Authors · 2022-11-18
> **Response to Reviewer Qimc**
>
> Thank you for your review. Please see below point-by-point responses to your comments.  We addressed common issues among reviewers [here](https://openreview.net/forum?id=p48GR3rwtxf&noteId=4aO9n1qvHy).
>
> > "*It remains however unconvincing as 1) the authors do not verify if their results hold across multiple runs with different random initializations of the networks or are only random artifacts*"
>
> Figures 2, 3, 5, 6, 7 and 8 show the statistics across 10 different runs with different initializations. We have added now this information in the appendix and in the figure captions.
>
> > "*The paper is clearly written, though the authors make undue allusions between their work and multi-task handling in the human brain*"
>
> The idea of studying different contexts at different processing stages for us comes from studies in neuroscience. We would appreciate if the reviewer would made the claim of "undue allusions between their work and multi-task handling in the human brain" more specific. An example or some sort of orientation about what is missing or "undue" would help us to fully address the issue.

---

### Official Review · Reviewer_px1r · 2022-10-25

**Confidence:** 4
**Correctness:** 3
**Technical Novelty And Significance:** 2
**Empirical Novelty And Significance:** 2
**Recommendation:** 3

**Clarity, Quality, Novelty And Reproducibility:**

The paper is clear but can benefit from proofreading. The quality of the analysis can be further improved as suggested above.

**Strength And Weaknesses:**

The paper studies an important problem which I believe it is not well studied yet. However, I have the following questions and comments for improving the paper:
- For the task switching network, having separate bias terms for each task makes sense but might be too simplistic. It would be interesting to do similar studies to what you already have with more flexible transformations such as the one proposed in Sun et al., 2021.
- minor: colorbar in Figure 2(a) seems to be incorrect (all red).
- Figure 3.1b in the text seems to refer to Figure 2.
- “switching networks, behave similarly as parallel in the first tasks” it should be in the first “layers”. Also the section is missing in “After observing the impact of context in the first layer (section ), “. Generally, the paper can benefit from some careful proofreading.
- The title for section 3.4 might be misleading as Figure 5.C doesn’t support that. Also I’m surprised that having a single layer with context is doing so well in terms of accuracy. This may be a hint that the task is too simple for an interesting analysis. A dataset with more diverse images may help in this experiment.
- How does the analysis change if done on a more challenging dataset like CIFAR-10 and also with a slightly more complicated network architecture like a CNN.
- I found some results to be interesting but it’s unclear how they can be used for designing better context-dependent multi-task learning algorithms.

**Summary Of The Paper:**

The paper investigates the effect of task-dependent context in multi-task learning for the task-switching networks. Particularly, the authors show the task-switching networks operate in a regime between two extreme cases of individual and parallel networks. The authors also study the shared representations, the role of context and its magnitude in simple multi-task learning experimental settings.

**Summary Of The Review:**

Overall, I found the exploratory analysis of context dependent multi-task learning an important analysis to be done. However, I believe the paper needs some work to either show the results hold in more general settings and difficult tasks, or provides some guideline on how to utilize the insights for improving multi-task learning algorithms.

---

> ### Author Response · Authors · 2022-11-18
> **Response to Reviewer px1r**
>
> Thank you for your review. Please see below point-by-point responses to your comments.  We addressed common issues among reviewers [here](https://openreview.net/forum?id=p48GR3rwtxf&noteId=4aO9n1qvHy).
>
> >"*For the task switching network, having separate bias terms for each task makes sense but might be too simplistic*"
>
> Adding the most minimalistic context input to the network made sense to avoid extra assumptions. More complicated versions of the context input would necessarily imply an underlying model-based version of the context input that could be seen both arbitrary and external to the internal representations of the network model. On the other hand, we agree that other internal mechanisms could also be tested, such as network modules dedicated to encode and process the task context. However, we think that simple models are better to begin with, to accumulate knowledge before tackling more complex scenarios. Together, these were the two main reasons why we analyzed neural representations being reshaped by context inputs through learnable weights.
>
> > "*minor: colorbar in Figure 2(a) seems to be incorrect (all red)*"
>
> The colorbar of Figure 2(a) is actually showing a transition from blue to gray to red. Is this perhaps a printer issue?
>
> > "*“switching networks, behave similarly as parallel in the first tasks” it should be in the first “layers”*"
>
> Thank you for catching this mistake. It has been corrected in the current version of the manuscript.
>
> > "*Also I’m surprised that having a single layer with context is doing so well in terms of accuracy*"
>
> In panels 5(a), 5(b) and 5(c) we can observe that the accuracy is almost similar for 10 layers, since the network has enough capacity (gray background denotes layers without context). This is confirmed by Figure 5(d) where one layer has significantly less accuracy (white background denotes absence of layer).
>
> > "*How does the analysis change if done on a more challenging dataset like CIFAR-10 and also with a slightly more complicated network architecture like a CNN*"
>
> Adding convolutions in the first layers would increase the complexity of the analysis. For instance tasks can be also encoded as extra channels in the input image, besides bias contexts in the dense layers. We think it is an interesting problem, but it's beyond the scope of this study.

---

### Official Review · Reviewer_opgD · 2022-10-25

**Confidence:** 4
**Correctness:** 2
**Technical Novelty And Significance:** 2
**Empirical Novelty And Significance:** 2
**Recommendation:** 3

**Clarity, Quality, Novelty And Reproducibility:**

Almost of figure numbers are incorrect. I should guess the matching figures.
A lot of typos make readers confusing.
A logical leap is seen between the result and the argument.
It is hard to agree with their proposes, I can't give much novelty.
The networks dimensions, optimizing parameters, regularization techniques, and other training hyper-parameters are not mentioned. difficult to reproduce without supplementary material.


**Strength And Weaknesses:**

Strengths
- Various analysis technique
- Well visualized results

Weaknesses
- Miss guided figure numbers make readers confusing. Correction is must needed, especially in the first half.
- I think tasks are too easy that switching-task might be not meaningful. As you showed in Figure 2(c), all models work well in both tasks. So Figure 2(a) has no meaning because outputs are too obvious. If you want to claim parameter sharing of networks, tasks should not be correlated, like the number of input and average value of input pixels.
- Why did you use one-hot vector as context vector? Adding not trainable one-hot vector as bias makes 'biased' networks. I want to see the activation values and its means. Dose adding 1 make too big changes in its output? If it is, the experiments about the context strengths is just come from the added bias, not the context. Using embedding vectors is reasonable choice if you care about the gap between value 1 and activation.
- In figure 5, you should show magnitude of individual network trained with one-hot vector used as bias to prove magnitude difference come from context.


**Summary Of The Paper:**

This paper compares three network architectures for multi-task learning. The experiment uses the MNIST dataset for two binary classification tasks. The authors categorized the output activation of each layer according to their input number to show their (un)correlation with the task. The authors also analyzed the structure of the task-switching network by changing the additional method or location, order, or strength of context vectors. Based on the results of these experiment and various analyzing methods, they present special features of task-switching networks. This paper proposes that adding context helps the network to generate expressive features and map tasks to appropriate outputs.

**Summary Of The Review:**

The authors did many experiments. But experimental results and their findings is not strongly connected.
The tasks they defined are highly correlated, so it is not possible to be sure that the results came from the context or the task, which is insufficient to supporting their findings.
Too many typos make uncomfortable experience.

---

> ### Author Response · Authors · 2022-11-18
> **Response to Reviewer opgD**
>
> Thank you for your review. Please see below point-by-point responses to your comments.  We addressed common issues among reviewers [here](https://openreview.net/forum?id=p48GR3rwtxf&noteId=4aO9n1qvHy).
>
> >"*Miss guided figure numbers make readers confusing. Correction is must needed, especially in the first half*"
>
> We have addressed this issue.
>
> > "*As you showed in Figure 2(c), all models work well in both tasks. So Figure 2(a) has no meaning because outputs are too obvious*"
>
> We fail to understand this statement. The point of the figure is precisely to show that good performance does not map onto specific neural selectivity. Despite showing that many different classifiers achieve similar overall performance, the internal neural representations are very different in each model, and their pattern was not something that we could anticipate.
>
> > "*Why did you use one-hot vector as context vector? Adding not trainable one-hot vector as bias makes 'biased' networks*"
>
> We are not sure we understand the comment. We use a one-hot vector to encode the task context. However, task context inputs to neurons are modulated by weights, so biases are learned through training.
>
> > "*In figure 5, you should show magnitude of individual network trained with one-hot vector used as bias to prove magnitude difference come from context*"
>
> We agree and added Figure 9 to show that the magnitude of the biases in individual networks are different from that of task-switching networks.
>
> > "*A logical leap is seen between the result and the argument. It is hard to agree with their proposes, I can't give much novelty*"
>
> Could you please elaborate? Without being specific, we cannot address the potential issues or give a proper response. Please explain the logical leap and what you mean by "our proposes" that is hard to agree with. Also, we think it is difficult to evaluate novelty if, to the opinion of the reviewer, there are logical leaps and "not agreeable" proposes.
>
> >"*The networks dimensions, optimizing parameters, regularization techniques, and other training hyper-parameters are not mentioned. difficult to reproduce without supplementary material*"
>
> In the original submission, we detailed all the information you see missing in the appendix and referenced in section 2. Also, we provided a zip file with the code to reproduce all the figures, which was mentioned in the reproducibility statement section after conclusions.

---

### Official Review · Reviewer_W7yc · 2022-10-26

**Confidence:** 3
**Correctness:** 3
**Technical Novelty And Significance:** 2
**Empirical Novelty And Significance:** 2
**Recommendation:** 3

**Clarity, Quality, Novelty And Reproducibility:**

The clarity and organization of this paper needs to be improved. Quality and technical novelty of this paper is relatively limited.

**Strength And Weaknesses:**

Strengths
- This paper studies a very meaningful research problem of understanding the representation learned for multi-task models.
- The authors utilized different statistical tools to conduct analyses on understanding the representation learned and importance of context for different multi-task learning model architectures.

Weaknesses
- Presentation. The presentation and organization of this paper can be improved. I found it hard to connect different analyses done in section 3. It might be good to have an overview of different research hypotheses or questions being answered in each of the analyses.

- Novelty. The technical novelty of this paper is relatively limited. The model architectures used in this paper to conduct analysis don’t reflect recent advances in multi-task deep learning. Also the analyses being done here is similar to other existing work that studies the understanding of multi-task deep learning. I also want to see if the authors can propose some novel improved techniques based on their findings.

- Related work. This paper doesn’t include recent advances in multi-task deep learning, either in understanding MTL or MTL model architectures, which limits its contribution and significance. For example, [1] conduct analysis on multi-task deep learning for its generalization ability related to representation learning and model capacity. Also, besides the task switching network, Mixture-of-experts [2,3] based approaches, and attention modules [4] have been widely used for multi-task learning.

[1] https://arxiv.org/abs/2005.00944
[2] https://proceedings.neurips.cc/paper/2021/hash/f5ac21cd0ef1b88e9848571aeb53551a-Abstract.html
[3] https://dl.acm.org/doi/pdf/10.1145/3219819.3220007
[4] https://arxiv.org/abs/1910.10683

- Experiment dataset. The dataset used in conducting the experiment is quite small. Multi-task learning generalization benefits can be better evaluated in different scenarios, such as: (1) tasks have similar amounts of data, (2) some tasks have much fewer data (3) few-shot tasks. These scenarios cannot be fully captured by MNIST dataset.


**Summary Of The Paper:**

This paper studies the problem of neural representation learning in multi-task learning. The authors compared three types of model architectures: individual network, parallel network and task switching network, on MNIST dataset with a multitask learning setting. To be more specific, the authors conducted analyses on the representations learned from these model architectures. In the discussion, the authors pointed out that task-switching networks can encode congruency to resolve multi-tasking, and the context of tasks can improve multi-task learning differently when being used at different positions (layer depth).

**Summary Of The Review:**

This paper conducted experiments and analyses on understanding neural representations in multi-task learning networks. There are some very interesting results and discussions. However, I think this paper can be further improved by including more recent advances in multi-task deep learning and discussion related to their findings. Meanwhile, the experiment section can be improved by including larger benchmark datasets.

---

> ### Author Response · Authors · 2022-11-18
> **Response to Reviewer W7yc**
>
> Thank you for your review. Please see below point-by-point responses to your comments.  We addressed common issues among reviewers [here](https://openreview.net/forum?id=p48GR3rwtxf&noteId=4aO9n1qvHy).
>
> > "*Related  work. This paper doesn’t include recent advances in multi-task deep  learning, either in understanding MTL or MTL model architectures, which  limits its contribution and significance. For example, [1] conduct  analysis on multi-task deep learning for its generalization ability  related to representation learning and model capacity. Also, besides the task switching network, Mixture-of-experts [2,3] based approaches, and  attention modules [4] have been widely used for multi-task learning.*
> > [1] https://arxiv.org/abs/2005.00944 [2] https://proceedings.neurips.cc/paper/2021/hash/f5ac21cd0ef1b88e9848571aeb53551a-Abstract.html [3] https://dl.acm.org/doi/pdf/10.1145/3219819.3220007 [4] https://arxiv.org/abs/1910.10683"
>
> Thank you for the references.  This line of research provides different techniques of sharing representations in MTL. Yet, we fail to see a similar analysis of neural representation in any of them. To our knowledge, the procedure used in our manuscript to analyze neural representations is novel in machine learning, since the study of mixed selectivity comes from neuroscience.
>
> > "*Multi-task learning generalization benefits can be better evaluated in different scenarios*"
>
> As mentioned earlier, our procedure is now ready to be applied to more sophisticated architectures in future studies conducted by us and, hopefully, by others as well.
>
> -----------------------------------------------------------------------------------------------------------------------------------------------------------
>
> ###

---

### Author Response · Authors · 2022-11-18
**Response to the Reviewers**

We thank the reviewers for their time and feedback. We apologize for the presence of typos and for using the default reference to figures that map them to section numbers instead of to the figures themselves. The new version of the manuscript fixes these issues.

There are some general comment by the reviewers that, perhaps unconsciously, aim to shift the intention of the study. We feel that we need to clarify this generally so that the purpose of the manuscript is made clear. The criticism can be summarized in three aspects: **benchmarking**, **simplicity** and **correlation**.

### Benchmarking

Benchmarking is out of the scope of our study. The research goal is on learning representations, i.e., to better understand the emergence of neurons' mixed selectivity in a neural network engaged in context-dependent decision making, rather than devising a deep learning model aimed at beating, frequently by a marginal margin, the current state of the art when tested in an arbitrary task, or set of tasks.

### Simplicity

Our goal underlies the reason of preferring simple models, as it makes the analysis and interpretation of mixed selectivity feasible, as well as it allows understanding how the shape of mixed selectivity is modulated through layers and by the context information. We work at the intersection between neuroscience and machine learning. To make the analogy, one cannot easily understand the brain by making models of the brain. Instead, neuroscientists need to cover all different levels of abstraction and when devising a model, its complexity needs to be set  according to the question or goal of the study. These does not mean that we, particularly, are not interested in studying more sophisticated models. However, we consider that we need to build up knowledge first, and to our opinion, our study now sets the basis to apply the same procedure to other more modular architectures.

### Correlation

Task correlation is in fact an essential ingredient in our experimental design. The analysis shown in Figure 2 does not make sense for uncorrelated tasks, as the result is obvious and can be easily anticipated: no systematic correlation of the neural activations between the two tasks across the 10 different realizations, exactly as it happens for the cross-correlations between the two individual networks in Figure 2(a), that is the two big, gray squares located in the anti-diagonal. The interesting and novel aspect that this analysis on correlated task brings is the possibility to demonstrate distinct mixed selectivity patterns between parallel networks and task-switching networks, something that is not obvious and we did not foresee in advance. Thus, for individual tasks, neural representations not surprisingly correspond to even/odd classification and higher/lower value, respectively (Figure 2(a), first row). Simultaneous parallel processing of the two tasks, Figure 2(a) second row, resolves context-dependent decision making by superseding either task representation of individual networks with the overlap of the two. We agree that a clever human being might come with such a solution after careful thinking. What is more far from obvious is the pattern evolution in the task switching network across layers and its final solution, Figure 2(a), third row. The processing of the first layers resemble that of the parallel network, only to converge in later layers (layer 5 here) to the representation of individual networks in its main diagonal, while in its anti-diagonal a completely new pattern emerges that is associated with the stimulus congruency, an aspect of context-dependent decision making to which we refer in the manuscript that is typically studied in psychology and behavioral neuroscience.

---

### Decision · Program_Chairs · 2023-01-20

**Decision:**

Reject

**Justification For Why Not Higher Score:**

The paper got straight 3's and was unconvincing to any of the reviewers

**Justification For Why Not Lower Score:**

N/A

**Metareview: Summary, Strengths And Weaknesses:**

This paper analyzes neural multi-task learning from a representation perspective. The authors compare three types of networks: individual (separate weights for each task), parallel networks (all weights but output layer shared), and task-switching networks (all weights but context vectors shared) using analyses such as RSA (Representational Similarity Analysis). They show that task-switching networks can encode congruency to resolve multi-tasking, and task context can improve multi-task learning when provided at different layers.

Strengths: Clearly this is an interesting and important problem and the authors have provided some insight in their analysis. The analyses presented are novel in this context as far as I can tell.

Weaknesses: Aside from typos and clarity which I assume the authors have addressed in their revision, there were two main weaknesses brought up: (1) lack of engagement with the large body of MTL literature developed since Caruana (see below for another reference); (2) too narrow and descriptive an analysis of too simple a model.

I understand the author's perspective on starting simple where it's easier to tease out and control the relevant factors. But the conclusion, for example, that weight sharing improves transfer, has been widely discussed and analyzed elsewhere and felt a bit unsurprising to the reviewers. The analysis is interesting but perhaps lacks some generality and is a bit narrowly focused on this particular setting.

Another reference https://arxiv.org/pdf/1711.01239.pdf.


**Summary Of Ac-Reviewer Meeting:**

N/A